# Pollen as Bee Medicine: Is Prevention Better than Cure?

**DOI:** 10.3390/biology12040497

**Published:** 2023-03-24

**Authors:** Maryse Vanderplanck, Lucie Marin, Denis Michez, Antoine Gekière

**Affiliations:** 1CEFE, Univ Montpellier, CNRS, EPHE, IRD, 34090 Montpellier, France; 2Laboratory of Zoology, Research Institute for Biosciences, University of Mons, 7000 Mons, Belgium

**Keywords:** environmental stressors, prophylaxis, therapy, tolerance, resistance, self-medication

## Abstract

**Simple Summary:**

Bees suffer from diverse pathogens and parasites that play crucial roles in shaping their communities. Alas, human activities have deeply disturbed natural bee–pathogen dynamics, through the spread of emerging pathogens and altered transmission networks. Such human-disturbed pollinator–pathogen dynamics are partly assumed to be responsible for pollinator decline. To deal with parasite infection, bees may rely on specific resources that may act as natural pharmacies. In this study, we explored whether different pollen resources may impede parasite establishment in healthy bumble bees or reduce parasite load in infected ones, either by weakening the parasite or improving the host resistance towards infection. Moreover, we also investigated whether infected bumble bees favour medicating resources over non-medicating ones. We found that consuming specific resources could impede parasite dynamics, especially its establishment, but that the cost–benefit trade-off could be detrimental if bumble bee reproductive success is highly reduced along with these medicinal effects. We did not affirm any self-medicative behaviour in infected bumble bees regarding the gut parasite used in our study, but questioned the importance of parasite virulence for such behaviours to occur. Our results showed that the ecological significance of medicating resource may be overlooked when missing parts of the story.

**Abstract:**

To face environmental stressors such as infection, animals may display behavioural plasticity to improve their physiological status through ingestion of specific food. In bees, the significance of medicating pollen may be limited by their ability to exploit it. Until now, studies have focused on the medicinal effects of pollen and nectar after forced-feeding experiments, overlooking spontaneous intake. Here, we explored the medicinal effects of different pollen on *Bombus terrestris* workers infected by the gut parasite *Crithidia bombi*. First, we used a forced-feeding experimental design allowing for the distinction between prophylactic and therapeutic effects of pollen, considering host tolerance and resistance. Then, we assessed whether bumble bees favoured medicating resources when infected to demonstrate potential self-medicative behaviour. We found that infected bumble bees had a lower fitness but higher resistance when forced to consume sunflower or heather pollen, and that infection dynamics was more gradual in therapeutic treatments. When given the choice between resources, infected workers did not target medicating pollen, nor did they consume more medicating pollen than uninfected ones. These results emphasize that the access to medicating resources could impede parasite dynamics, but that the cost–benefit trade-off could be detrimental when fitness is highly reduced.

## 1. Introduction

To protect themselves from structural and functional damage (e.g., protein oxidation, organ dysfunction) due to external challenges, organisms have evolved complex machineries of endogenous defences and physiological adjustments [1,2]. These physiological responses are involved in many essential pathways such as immune defence, cellular signalling, thermal tolerance and survival. However, mounting an appropriate physiological response to face environmental stressors may be inadequate, as it may take explicit time for activation [3], and it requires organisms to be in good conditions as it incurs a cost to recruit endogenous resources [4]. To alleviate the costs related to the use of endogenous resources, or to compensate for latency or for limited physiological plasticity, organisms have the possibility to modify their behaviour (i.e., Bogert effect, [5]), for instance by ingesting specific exogenous resources. Such a regulation of environmental stress using food items could be one of the key mechanisms determining the capacity of organisms to cope with external challenges commonly encountered in an ever-changing world.

Ingestion of pharmaceutically active substances may immediately correct for suboptimal physiological state as observed in *Bicyclus anynana* (Lepidoptera: Nymphalidae) exposed to high temperatures [6]. During the feeding experiment, this tropical butterfly adjusted its behaviour by doubling its polyphenol intake under hot conditions, avoiding oxidative damage without affecting its endogenous antioxidant defences. Such ingestions of food items for physiological improvement could be used not only therapeutically (i.e., after facing the environmental stress) but also prophylactically (i.e., before facing the environmental stress), with postponed advantages [7]. For instance, many bird species consume fruits rich in antioxidants before migration to reduce the oxidative stress induced by long flights [8]. However, the prophylactic use of pharmaceutically active substances implies to be able to store and rapidly mobilize them in response to an external challenge, which is not possible for all dietary components (e.g., hydrophylic antioxidants are not easily stored in the organism [9]). The ingestion of specific food items by organisms to improve their physiological and health status corresponds to the notion of “self-medication”, which occurs in many taxa [7,10,11,12,13,14]. Following an evolutionary framework, the conditions necessary for self-medication against environmental stress through the consumption of specific food items are (i) the frequent exposure to environmental stressors over life and evolutionary times, (ii) the evolution in parallel with suitable medicating resources, (iii) the possibility for organisms to recognise medicating resources, (iv) the alleviation of the cost associated with the use of endogenous defences, (v) the cost of the consumption of medicating resources in the absence of environmental stress and (vi) the alleviation of fitness loss caused by environmental stress [15]. Bees (Hymenoptera: Anthophila) constitute an excellent model to investigate the potential for self-medication to face global changes as they are exposed to multiple environmental stressors, experience population declines [16,17], and share a long evolutionary history with their nutritional resource (i.e., flowering plants [18,19]).

Hitherto, studies addressing potential self-medicative behaviours in bees are scarce and mainly concern eusocial corbiculate species, including honey bees and bumble bees. Simone-Finstrom and Spivak [20] showed that honey bees increased resin collection when challenged with chalkbrood disease, but demonstrating resin-induced costs at the colony level still remains a major hurdle [21]. Gherman et al. [22] and Ferguson et al. [23] showed that *Nosema*-infected honey bees preferentially foraged on specific resources but failed to find any detrimental effect of the latter on uninfected bees. In bumble bees, Baracchi et al. [24] observed the hampering effect of nicotine towards a gut parasite and demonstrated the bumble bee preference for nicotine-laced solution. However, they could not observe any detrimental consequences due to the infection. Therefore, to date, no study has properly demonstrated any case of self-medication in bees. In the present study, we focused on the prophylactic and therapeutic effects of different pollen diets on parasitized bumble bees using forced-feeding assays (experiment on medicinal effects). We also looked for spontaneous pollen intake when given a choice between food items to assess the ability of parasitized bumble bees to exploit medicating pollen (experiment on self-medication). In the context of a parasitic stress, the definition of medicating resources, and specifically of medicinal effects themselves, may be confusing. Indeed, a medicinal effect can occur either by benefiting the host (i.e., increased host tolerance) or by harming the parasite (i.e., increased host resistance). While some have claimed that a food item must be detrimental to the parasite to be considered as medicinal [25], recent studies have argued that it is not mandatory and proposed that medicinal effects could either increase host resistance or tolerance to infection [26]. We applied this second definition in this study and followed a three-step approach to assess the importance of medicating pollen for the bumble bee *Bombus terrestris* submitted to a parasitic stress (see Appendix B for description of model species): using different pollen species, we (i) assessed their prophylactic and therapeutic effects on fitness components (i.e., host tolerance), (ii) assessed their prophylactic and therapeutic effects on parasite loads (i.e., host resistance) and (iii) examined whether bumble bees favoured medicating pollen species when they could benefit the most from them (i.e., self-medication).

## 2. Materials and Methods

### 2.1. Prophylactic and Therapeutic Effects of Pollen

#### 2.1.1. Experimental Design

Medicinal effects of pollen (i.e., prophylactic and therapeutic ones) were investigated using *B. terrestris* microcolonies, following the method adapted from Regali and Rasmont [27]. Microcolonies consist of five two-day-old workers of *B. terrestris* without a queen placed in a plastic box (10 × 16 × 16 cm), whereby one worker will lay unfertilized male-destined eggs. Herein, they were established using three different founding queenright colonies equally distributed among 11 different treatments to ensure homogeneity (n = 9 microcolonies per treatment): (i) uninfected control, which consists in uninfected microcolonies fed with a multifloral diet (i.e., 25% willow, 25% sunflower, 25% orchard, 25% heather); (ii) prophylactic multifloral, which consists in infected microcolonies fed with a multifloral diet before worker inoculation; (iii) therapeutic multifloral, which consists in infected microcolonies fed with a multifloral diet after worker inoculation; (iv) prophylactic willow, which consists in infected microcolonies fed with a willow diet before worker inoculation; (v) therapeutic willow, which consists in infected microcolonies fed with a willow diet after worker inoculation; (vi) prophylactic sunflower, which consists in infected microcolonies fed with a monofloral sunflower diet before worker inoculation; (vii) therapeutic sunflower, which consists in infected microcolonies fed with a monofloral sunflower diet after worker inoculation; (viii) prophylactic poppy, which consists in infected microcolonies fed with a monofloral poppy diet before worker inoculation; (ix) therapeutic poppy, which consists in infected microcolonies fed with a monofloral poppy diet after worker inoculation; (x) prophylactic heather, which consists in infected microcolonies fed with a monofloral heather diet before worker inoculation and (xi) therapeutic heather, which consists in infected microcolonies fed with a monofloral heather diet after worker inoculation (Figure 1). The multifloral diet has been used as the control as it better reflected the generalist behaviour of *B. terrestris*. Microcolonies were fed *ad libitum* with sugar syrup (water:sugar 35:65 *w*/*w*) and maintained in a dark room at 26 ± 1 °C and 60 ± 10% relative humidity. After a four-day initiation phase (i.e., syrup only), pollen was provided as candies (i.e., pollen mixed with water and syrup). Microcolonies were then fed with pollen and syrup for a 15-day period (i.e., pollen feeding phase), and pollen candies were freshly prepared and renewed every two days (1–2 g depending on the age of the microcolony) to avoid nutrient alteration and drying out during the experiment. In prophylactic treatments, workers were inoculated individually with *C. bombi* during the pollen feeding phase (i.e., one day after the start of pollen feeding phase); while in therapeutic treatments, workers were inoculated individually with *C. bombi* during the initiation phase (i.e., two days before the start of pollen feeding phase) (Figure. 1) (see Appendix C for description of parasite reservoirs and inoculation). To avoid any handling bias, uninfected workers (i.e., uninfected control) were also isolated and starved before being placed in their respective microcolony. Workers that died during the experiment were weighed and replaced by new workers originating from the same foundress colony. The replacing workers did not undergo initiation phase or inoculation and were marked (coloured dot) to avoid any bias in measured parameters. Syrup and pollen supplies as well as microcolonies monitoring were conducted under red light in order to avoid any disturbance, as bees do not detect this range of the light spectrum. The experiments were conducted at the University of Mons from March to April 2022.

#### 2.1.2. Host Tolerance

Alleviation of costs associated with the parasite challenge (i.e., fitness loss, endogenous defenses) through the consumption of medicinal pollen can be identified by evaluating modifications in microcolony growth (i.e., brood mass), individual immunocompetence (i.e., worker fat body content and mortality), as well as in worker behaviour. For instance, the parasite could alter worker behaviour by increasing larval ejection from the brood. Larval ejection is a common process carried out by workers, and sometimes queens, in bumble bee colonies. This biological mechanism involves pulling live larvae out of cells and depositing them outside of the nest, which can control the number of larvae within the brood. This behavior is not well studied but appears to occur when the colony is under stress such as resource deficiency (i.e., stress response) [28,29,30,31]. Another stress response was evaluated through pollen efficiency (i.e., the mass of hatched offspring divided by the mass of collected pollen per microcolony), which highlights when workers need to collect more pollen to produce offspring, and could then be indicative of a resource allocation constraint by directing energy to immune responses. To assess these parameters, pollen collection was measured every two days during the experiment by weighing pollen candies before their introduction into the microcolony and after their removal. Pollen collection data were corrected for evaporation. Besides, dead workers and ejected larvae were counted and removed every day during the pollen feeding phase. At the end of the experiment, workers were weighed and their abdominal fat body content was measured as an indicator of immunocompetence (i.e., two workers per microcolony; [32]) using Ellers’ procedure [33,34]. The brood was dissected for recording the number and mass of individuals within each developmental stage (i.e., eggs, non-isolated larvae, isolated and pre-defecating larvae, isolated and post-defecating larvae). The brood mass was standardised by the total weight of workers in the microcolonies to avoid potential bias from worker activities.

To test for differences in microcolony growth, individual immunocompetence and worker behaviour, we fitted general linear mixed effect models (GLMMs) with treatments (Pollen*Inoculation, 11 levels) as a fixed effect and colony as a random factor. Brood mass and pollen efficiency per microcolony were analysed using models with a Gaussian error structure (i.e., normally distributed residuals, “lmer” function, R-package “lmerTest” [35]). Larval ejection and worker mortality were analyzed using a binomial model with the number of failures (i.e., ejected larvae or dead workers) and the number of successes (i.e., total number of living offspring produced per microcolony or total number of living workers per microcolony) as a bivariate response (“glmer” function, R-package “lmerTest” [35]). Because complete separation occurs for larval ejection data (i.e., Hauck-Donner effect), we used a Bayesian inference by setting a default prior (i.e., zero-mean Normal) on the fixed effect (“bglmer” function, R-package “blme” [36]). Fat body content was analysed using models with a Gamma distribution error structure and a logit link to deal with proportion data. For each parameter, contrasts were then performed on the models to determine whether treatments differed from the uninfected control, whether prophylactic or therapeutic effects differed among pollen, and whether the medicinal effect of pollen differed according to the inoculation (prophylactic vs therapeutic) (“contrast” function, R-package “emmeans” [37]). All analyses were performed in R version 4.2.0.

#### 2.1.3. Host Resistance

Reduction of infection (i.e., therapeutic effect) or loss of infectivity (i.e., prophylactic effect) through the consumption of pollen was assessed by monitoring the parasite load on one marked worker during the experiment. The first measure was taken three days after inoculation, as soon as infection is known to turn patent [38], and following measures were then taken every three days (Figure 1). The marked worker was placed in a 50 mL Falcon tube, faeces were collected using a 10 µL microcapillary and the volume was measured. The faecal sample was then diluted (i.e., dilution 5× or 10× according to the load) to allow for counting the *C. bombi* cells by using an improved Neubauer haemocytometer at 400-fold magnification under an inverted phase contrast microscope (Eclipse Ts2R, Nikon; Tokyo, Japan). Only flagellated stages were considered to evaluate the parasite load. Despite the possibility for death of marked workers during the experiment (one marker worker died in the multifloral prophylactic treatment at the end of the experiment) or for absence of faeces (three non-defecating events were registered), nine replicates were mostly available at each time point for all treatments (eight replicates in the four cases above-mentioned).

We used generalized additive mixed-effect models (GAMMs) to compare parasite loads over time for all treatments with pollen and inoculation included as crossed fixed effects, faeces volume included as covariate and microcolony included as random factor. Data were log-transformed and two outliers were discarded to achieve normal distribution. We additionally assessed differences in the parasite establishment (i.e., initial load at day 4) and growth among treatments using GLMMs with pollen and inoculation included as crossed fixed effects, and colony as random factor (Gaussian error structure, log-transformed data). The parasite growth was defined as the change in parasite load (i.e., difference between final and initial measures) divided by the initial load. Contrasts were then performed on the models to determine whether prophylactic or therapeutic effects differed among pollen, and whether medicinal effect of pollen differed according to the bioassay (prophylactic vs therapeutic) (“contrast “function, R-package “emmeans” [37]). All analyses were performed in R version 4.2.0.

### 2.2. Self-Medicative Behaviour

#### 2.2.1. Experimental Design

Based on results from the first experiment (i.e., therapeutic pollen), potential for self-medicative behaviour in *B. terrestris* was evaluated using a feeding choice experiment with infected and uninfected microcolonies. Each microcolony consisted of five workers placed in a two-box system (10 × 16 × 16 cm per box) maintained at 26 ± 1 °C and 60 ± 10% relative humidity. Food (i.e., pollen and syrup) was provided in one box kept under 12 h natural light conditions (08:00–20:00) and divided in two compartments for the pollen choice (feeding chamber), while the other box was kept in dark conditions and housed the brood (nest chamber). The two boxes communicated through a flexible transparent tube (10 cm) with a diameter enabling worker passage (1.6 cm) (Figure 2). Uninfected and infected microcolonies were offered one out of three different pollen combinations, namely heather/multifloral, sunflower/multifloral and heather/sunflower, which resulted in six treatments. Nine queenless microcolonies were established for each treatment using two-day-old workers of *B. terrestris* from three different queenright colonies equally distributed among treatments to ensure homogeneity. After a four-day initiation phase (i.e., syrup in the feeding chamber and a small willow candy in the nest chamber for brood initiation), pollen was provided ad libitum as candies (i.e., pollen mixed with water and syrup) in the feeding chamber, and microcolonies were fed with pollen (i.e., binary choice) and syrup for an eight-day period (i.e., pollen feeding phase). Every three days, we provided fresh pollen candies in small caps in the two compartments of the feeding chamber. In infected treatments, workers were inoculated individually with *C. bombi* during the initiation phase (i.e., two days before the onset of the pollen feeding phase). Workers that died during the experiment were removed but not replaced as pollen collection data were paired within treatment. The experiments were conducted at the University of Mons from late June to early July 2022.

#### 2.2.2. Pollen Preference and Infection Cure

Caps were weighed daily to measure pollen uptake by each microcolony, and data were corrected for evaporation using controls. The position of caps was randomised daily to prevent any positional bias. To test for self-medicative behaviour, we used GAMMs to compare daily amounts of pollen collected from each pollen between infected and uninfected microcolonies, with microcolony included as random factor (Gaussian error structure). Data were processed separately for each pollen combination (i.e., three GAMMs). Additionally, we compared the total pollen amount that each microcolony collected from each of the two diets offered within each treatment (i.e., infection*pollen combination) using paired *t*-tests after checking for normality. We then tested for the difference in pollen preference between infected and uninfected microcolonies using GLMMs with infection status as a fixed effect, and colony as a random factor (R-package “lmerTest” [35]). We used a binomial model with the consumption of both pollen choices as a bivariate response after checking for overdispersion. Data were processed separately for each pollen combination (i.e., three GLMMs). Moreover, the effects on infection were assessed by measuring the parasite load on two marked workers per microcolony three days after inoculation, and at the end of the experiment. When both marked workers survived, one was randomly selected to be included in the analyses; while the survivor was automatically considered when one died during the experiment. There has never been the case of the two marked workers dead so that nine replicates were available for all treatments. We assessed differences in the parasite establishment and growth among treatments using GLMMs with pollen choice included as fixed effects, and microcolony nested in colony as random factor (Gaussian error structure, log-transformed data). All analyses were performed in R version 4.2.0.

## 3. Results

### 3.1. Prophylactic and Therapeutic Effects of Pollen

#### 3.1.1. Host Tolerance

Regarding the medicinal effects of pollen on the microcolony growth, the brood mass of uninfected microcolonies was higher than for all therapeutic treatments, and significantly differed from microcolonies fed sunflower pollen (*t* = −4.437, *p* < 0.001). In prophylactic treatments, infected microcolonies fed multifloral pollen displayed the highest growth, even compared to the uninfected microcolonies, and differed significantly from microcolonies fed poppy pollen (*t* = 2.928, *p* = 0.004); heather pollen (*t* = 2.715, *p* = 0.008); and sunflower pollen (*t* = 5.688, *p* < 0.001). By contrast, brood mass of microcolonies fed sunflower pollen was the lowest for both therapeutic (significantly different from the uninfected control, *t* = −4.437, *p* < 0.001; and microcolonies fed willow pollen, *t* = 3.059, *p* = 0.003; and multifloral pollen, *t* = 2.858, *p* = 0.005) and prophylactic treatments (significantly different from the uninfected control, *t* = −3.831, *p* < 0.001; and microcolonies fed other pollen, *p* < 0.01). The only significant difference detected between therapeutic and prophylactic treatments was in microcolonies fed multifloral pollen (*t* = 3.437, *p* < 0.001), those in prophylactic treatments displaying a higher growth (Figure 3, Appendix A).

Regarding worker behaviour and stress responses, pollen efficiency in uninfected microcolonies was the highest in therapeutic treatments (significantly different from microcolonies fed sunflower pollen, *t* = −4.928, *p* < 0.001) while infected microcolonies fed multifloral pollen displayed the highest growth in prophylactic bioassays (significantly different from microcolonies fed sunflower pollen, *t* = 5.004, *p* < 0.001). By contrast, pollen efficiency in microcolonies fed sunflower pollen was the lowest for both therapeutic (significantly different from the uninfected control, *t* = −4.928, *p* < 0.001; and microcolonies fed other pollen, *p* < 0.01) and prophylactic treatments (significantly different from the uninfected control, *t* = −4.015, *p* < 0.001; and microcolonies fed other pollen, *p* < 0.01). The only significant difference detected between therapeutic and prophylactic treatments was in microcolonies fed multifloral pollen (*t* = 2.454, *p* = 0.016), those in prophylactic treatments displaying a higher pollen efficiency (Figure 4C, Appendix A). Regarding the brood, larval ejection was the highest within microcolonies fed sunflower pollen in therapeutic treatments (significantly different from the uninfected control, *t* = 4.597, *p* < 0.001; and from microcolonies fed other pollen, *p* < 0.001), and in prophylactic treatments (significantly different from the uninfected control, *t* = 4.292, *p* < 0.001; and from microcolonies fed other pollen, *p* < 0.001). Microcolonies fed heather pollen also displayed higher larval ejection compared to the uninfected control (*t* = 3.045, *p* = 0.002), and to infected microcolonies fed multifloral pollen (*t* = −3.016, *p* = 0.003), poppy pollen (*t* = −2.687, *p* = 0.007), and willow pollen (*t* = −2.800, *p* = 0.005) but only in prophylactic bioassays. No significant difference was detected between therapeutic and prophylactic treatments, irrespective of the pollen diets (Figure 4A, Appendix A).

#### 3.1.2. Host Resistance

The infection dynamic was more gradual in therapeutic treatments compared to prophylactic ones (i.e., lower increase), with a significant difference between treatments for willow pollen (*t* = 3.827, *p* < 0.001), sunflower pollen (*t* = 4.949, *p* < 0.001), poppy pollen (*t* = 3.319, *p* = 0.001), and heather pollen (*t* = 3.666, *p* < 0.001) (Figure 5). Within both therapeutic and prophylactic treatments, the infection was significantly hampered in microcolonies fed heather and sunflower pollen compared to those fed either willow, poppy or multifloral pollen (Therapeutic: heather vs. willow, *t* = 4.228, *p* < 0.001; heather vs. poppy, *t* = 4.760, *p* < 0.001; heather vs. multifloral, *t* = 5.614, *p* < 0.001; sunflower vs. willow, *t* = 6.837, *p* < 0.001; sunflower vs. poppy, *t* = −7.408, *p* < 0.001; sunflower vs. multifloral, *t* = 8.276, *p* < 0.001; Prophylactic: heather vs. willow, *t* = 4.408, *p* < 0.001; heather vs. poppy, *t* = 4.401, *p* < 0.001; heather vs. multifloral, *t* = 3.888, *p* < 0.001; sunflower vs. willow, *t* = 5.784, *p* < 0.001; sunflower vs. poppy, *t* = −5.722, *p* < 0.001; sunflower vs. multifloral, *t* = 5.201, *p* < 0.001) (Figure 5, Appendix A). While sunflower and heather pollen displayed the same effect on the infection dynamic during the prophylactic treatments (*t* = −1.332, *p* = 0.184), the hampering effect of sunflower pollen on the infection dynamic was stronger than that of heather pollen during the therapeutic treatments (*t* = −2.557, *p* = 0.011) (Figure 5, Appendix A).

When dissecting the influence of pollen on host resistance, parasite establishment appeared significantly more hindered in therapeutic treatments compared to the prophylactic ones (willow, *t* = 3.434, *p* = 0.001; sunflower, *t* = 6.139, *p* < 0.001; poppy, *t* = 2.042, *p* = 0.045; and heather, *t* = 4.552, *p* < 0.001), except for the multifloral pollen (*t* = 1.088, *p* = 0.280) (Figure 6A, Appendix A). In both prophylactic and therapeutic treatments, parasite reached higher densities at the beginning of infection in microcolonies fed poppy pollen while establishment was more impeded in microcolonies fed sunflower pollen (Figure 6A, Appendix A). By contrast, parasite growth was higher in therapeutic treatments compared to the prophylactic ones with a significant difference in microcolonies fed heather pollen (*t* = −2.709, *p* = 0.009). While the parasite displayed the lowest growth in microcolonies fed poppy pollen irrespective of the treatments (Therapeutic: poppy vs. willow, *t* = 2.546, *p* = 0.013; poppy vs. sunflower, *t* = 2.595, *p* = 0.012; poppy vs. heather, *t* = −3.108, *p* = 0.003; poppy vs. multifloral, *t* = 2.465, *p* = 0.016; Prophylactic: poppy vs. multifloral, *t* = 4.183, *p* < 0.001), the highest growth was observed in microcolonies fed heather pollen in therapeutic treatments (heather vs. poppy, *t* = −3.108, *p* = 0.003) and in microcolonies fed multifloral pollen in prophylactic treatments (multifloral vs. willow, *t* = −2.615, *p* = 0.011; multifloral vs. sunflower, *t* = 2.732, *p* = 0.008; multifloral vs. poppy, *t* = 4.183, *p* < 0.001; multifloral vs. heather, *t* = 3.142, *p* = 0.003) (Figure 6B, Appendix A).

### 3.2. Self-Medicative Behaviour

Regardless of their infection status, bumble bee workers collected more sunflower pollen (i.e., higher daily amount) than heather pollen (Infected, *t* = −2.23, *p* = 0.027; Uninfected, *t* = −3.09, *p* = 0.002) and more multifloral pollen than either sunflower (Infected, *t* = 2.65, *p* = 0.009; Uninfected, *t* = 2.88, *p* = 0.004) or heather pollen (Infected, *t* = −9.32, *p* < 0.001; Uninfected, *t* = −6.83, *p* < 0.001) (Figure 7, Appendix A).

Overall, pollen preference of bumble bee workers did not change according to their infection status (heather vs. multifloral, χ^2^ = 0.227, df = 1, *p* = 0.634; sunflower vs. multifloral, χ^2^ = 0.600, df = 1, *p* = 0.439; heather vs. sunflower, χ^2^ = 0.071, df = 1, *p* = 0.790), and bumble bee workers did not prefer medicinal pollen (heather or sunflower) over multifloral pollen (heather vs. multifloral, *t* = 4.157, df = 8, *p* = 0.003; sunflower vs. multifloral, *t* = 2.252, df = 8, *p* = 0.054), even when infected (heather vs. multifloral, *t* = 5.202, df = 8, *p* = 0.001; sunflower vs. multifloral, *t* = 1.974, df = 8, *p* = 0.084) (Figure 8A, Appendix A). Regarding pollen preference within treatments, bumble bee workers only preferred multifloral pollen over heather pollen, irrespective of their infection status (infected, heather vs. multifloral, *t* = 5.202, df = 8, *p* = 0.001; uninfected, *t* = 4.157, df = 8, *p* = 0.003). Besides, parasite establishment (χ^2^ = 5.49, df = 2, *p* = 0.064) and parasite growth (χ^2^ = 2.54, df = 2, *p* = 0.281) appeared similar between all treatments (Figure 8B).

## 4. Discussion

### 4.1. Is Prevention Better Than Cure?

#### 4.1.1. Pollen Diets Override Parasite Impacts

*Crithidia* infection (i.e., infected vs. uninfected microcolonies fed on multifloral pollen) did not significantly affect microcolony growth, individual immunocompetence or worker behaviour, as expected based on previous studies e.g., [39]. However, we found pollen species-dependant effects on host tolerance in infected microcolonies. Most notably, sunflower pollen led to the highest larval ejection, as well as to the lowest brood mass and pollen efficiency. Such detrimental effects of sunflower pollen on microcolony growth and worker behaviour have already been stressed in previous studies conducted on healthy bumble bees, and are likely due to a lack of essential amino acids, toxic specialised metabolites, and hardened exine [39,40,41]. Microcolonies fed heather pollen also displayed increased larval ejection compared to those fed other pollen diets (except sunflower), but it was significant only in the prophylactic treatments. The only differences observed between prophylactic and therapeutic treatments concerned the brood mass (microcolony growth) and the pollen efficiency (worker behaviour), which tended to be higher in prophylactic treatments compared to therapeutic ones, with significant effects in microcolonies fed the multifloral diet. As the feeding-inoculation sequence was the only methodological difference between prophylactic (i.e., feeding before inoculation) and therapeutic (i.e., inoculation before feeding) treatments, we propose three non-mutually exclusive rationales to explain these observations: (i) given that *C. bombi* likely require nutrients from pollen to develop [38], parasite establishment in therapeutic treatments may prevent the efficient use of pollen for brood building in the first days of feeding; (ii) given that *C. bombi* infection was found to reduce ovary size [42], parasite establishment in therapeutic treatments may have delayed egg laying; and (iii) given that *C. bombi* is known to be more virulent when combined with food deprivation [43,44,45], the concomitant nutritional and parasitic stresses could explain why bumble bee microcolonies showed reduced brood mass in therapeutic treatments.

Regarding individual immunocompetence and survival, the worker mortality in infected microcolonies did not differ from uninfected control, regardless of the inoculation or diet treatment. The only notable observation is that the worker mortality in infected microcolonies prophylactically fed willow pollen was quite high compared to the other treatments, differing significantly from infected microcolonies prophylactically fed heather pollen, and those therapeutically fed willow. Since laboratory alive parasite stocks used for the inoculation were kept by transmitting infection across commercial colonies only fed willow pollen for a year, we suggest that a selection of *C. bombi* strains well-suited for willow pollen may have inadvertently occurred. This selection would then have led to higher infection costs in prophylactic treatments with willow pollen. This hypothesis echoes the study of Palmer-Young et al. [46] who demonstrated such a selection for phytochemically resistant strains, and is supported by the higher parasite load three days after inoculation in microcolonies prophylactically fed willow pollen. However, it should be interpreted with caution, as it partly contrasts with the study of Palmer-Young et al. [47] who showed that *C. bombi* cells developing in vitro resistance towards the phytochemical eugenol did not have a higher infectiousness than naïve parasite cells in eugenol-fed bumble bees. Regarding fat body content, although it is commonly used as a proxy for individual immunocompetence e.g., [31,34,38,48], underlying mechanisms remain poorly understood, making difficult clear interpretations of any effect. In prophylactic treatments, poppy pollen led to slightly higher fat body content than sunflower and multifloral pollen, whereas heather pollen led to reduced fat body content when compared to any other diet in therapeutic treatments. These differences among diets could be partly explained by pollen central metabolites: while discussing around the multifloral pollen nutritional properties is not feasible since 25% of the mix is not utterly defined (i.e., orchard mix), both sunflower and heather pollen are characterized by the presence of peculiar sterols (i.e., delta-7 sterols) and lower amino acid concentrations compared, for instance, to poppy pollen [28,49,50,51]. However, although central metabolites likely play a role towards fat body content, the feeding-inoculation sequence seems to impact this metric as results differed between prophylactic and therapeutic treatments. Overall, we advocate further studies to disentangle the roles of pollen diet and feeding-inoculation sequence in influencing parasite strain selection and infectiousness.

#### 4.1.2. Specific Pollen Diets Hamper Parasite Establishment

Pollen diets impacted bumble bee worker resistance towards infection, as shown by differences in parasite loads and dynamics across treatments. More specifically, both sunflower and heather pollen impeded *Crithidia* development in both therapeutic and prophylactic treatments. While therapeutic effects of sunflower pollen have already been demonstrated in *Bombus impatiens* e.g., [52,53], it contrasts with a previous study conducted on *B. terrestris* [39]. This discrepancy could be explained by the difference between experiment durations. Indeed, when focusing on the first ten days post-inoculation in Gekière et al. [39], results appeared quite similar between experiments (i.e., significant reduction in parasite load). This highlights that effects of pollen diet on parasite load clearly vary according to the time span considered after inoculation, and that great caution around parasite load should be paid when studying medicinal effects. In both therapeutic and prophylactic treatments, our results highlighted that diets influence the parasite dynamic mainly at the beginning of infection, but have rather limited impacts once the infection is well-established. Actually, it seems that parasite load often plateaus or displays a chaotic pattern after a while (usually 10 days post-inoculation) [54,55], which has been observed for most treatments herein. For instance, both sunflower and heather pollen hampered parasite establishment compared to willow pollen but no difference was detected in parasite growth, and poppy pollen seemed to boost parasite establishment while growth index was slighter afterwards.

The most stunning results concern the discrepancies in the parasite load dynamic between prophylactic and therapeutic treatments: except for multifloral pollen, parasite establishment was reduced in workers fed therapeutically, resulting in a more gradual infection dynamic compared to workers fed prophylactically. This observation highlights the importance of distinguishing prophylaxis and therapy when dealing with infection and medicative behaviours. For instance, Koch et al. [56] found a preventive but not curative effect of the phytochemical callunene on *Crithidia*-infected bumble bees. They showed this molecule did not reach the hindgut where *Crithidia* cells principally establish and thus could not hamper existing infection. By contrast, they measured a high concentration of callunene in the foregut, damaging *Crithidia* cells before they headed for the hindgut, thereby preventing infection. Later, Koch et al. [57] demonstrated the opposite for the phytochemical tiliaside. This molecule did not damage *Crithidia* cells in vitro, or in the foregut, since it required to be deglycosylated during gut passage to become active against *Crithidia* in the hindgut, which highlighted the complexity and diversity of medicinal effects. Here, we propose a simpler explanation directly related to our experimental design, and more specifically to the feeding-inoculation sequence: whereas parasites directly had access to pollen through the whole gut upon inoculation in prophylactic treatments (i.e., feeding before inoculation), which likely favoured their development (i.e., parasite establishment [38,58]), parasites had to reach the hindgut without pollen in therapeutic treatments (i.e., inoculation before feeding), which likely reduced the number of establishing parasite cells. All these observations support the proposition that parasite establishment dictate the subsequent infection dynamic.

### 4.2. Are Infected Bumble Bees Able to Self-Medicate?

#### 4.2.1. The Importance of Parasite Virulence

During our choice experiment, bumble bee workers did not change their feeding behaviour according to their infection status. Such an absence of self-medicative behaviour is quite unexpected as bumble bees are known to display a huge plasticity in their feeding behaviours [59], and as self-medication has strongly been suggested for these species [24]. This might be due to an important limitation we pinpointed in our study. Actually, one condition for self-medication to occur against infection is the significant alleviation of host fitness costs induced by the parasite through the consumption of medicinal resource [15]. However, we did not observe any detrimental effect of *Crihidia* on bumble bee fitness under optimal breeding parameters, without flying and foraging activities, which prevents us from properly conclude about alleviation of fitness costs. For this condition to be met, we advocate the use of a more virulent parasite for bumble bees, such as *Apicystis bombi* [60], as done in the caterpillar *Spodoptera littoralis* by using a lethal parasite to demonstrate self-medicative behaviours (Lepidoptera: Noctuidae) [61]. Surprisingly, no therapeutic effect on host resistance has been observed in bumble bee workers that were offered the two medicinal resources (i.e., sunflower and heather pollen). This could be explained by the similar parasite establishment in all treatments, probably because parasite benefited from willow pollen provided in the nest chamber for establishing within the gut.

#### 4.2.2. The Neglected Role of Sociality

Since the initial self-medication framework formulated by Clayton and Wolfe [62], major changes have been adopted, notably with the concept of “kin” or “social medication” [26]. Indeed, it has been demonstrated that animals may medicate not only themselves but also their genetic kin, including offspring and other genetic relatives e.g., [63]. This adds a degree of complexity when investigating self-medicative behaviours since costs and benefits must be considered at both individual and social levels [21], especially in exclusively social species such as *B. terrestris* [64]. Although we used microcolonies to take into account this social issue in our experimental design, we may have missed social medicative behaviours that would have been observed in queenright colonies [65,66]. Moreover, as no brood clump has the time to be built in nest chambers, brood effect on worker feeding behaviour e.g., [67] was ruled out during our experiment, which prevents a fully complete picture of the self-medication. It would therefore be interesting to run future experiments using queenright colonies with well-developed brood, and allowing behaviours related to flying and foraging activities.

## 5. Conclusion and Future Directions

### 5.1. Designing an Adequate Control Diet

One limitation in our experimental design was the feeding-inoculation sequence that led to an inoculation right after a four-day “pollen starvation” period in therapeutic bioassays, resulting in an additional stress for infected bumble bees. One way to overcome this limitation would be to rely on a diet shift rather than on a starvation–feeding transition, by providing bumble bee microcolonies with a specialized metabolite-free and nutritionally well-balanced diet prior to feeding them with pollen candidates [39]. However, hitherto, researchers have failed to develop a specialized metabolite-free artificial diet which supports bumble bee colony growth [48,68]. Moreover, even such an artificial diet could lead to experimental limitation as central metabolites (i.e., carbohydrates, proteins and lipids) play pivotal roles in host–parasite interactions (e.g., see [69] for the importance of sugars for in vitro *Crithidia bombi* growth). Yet, we are not aware of any study tackling the relationships between in vivo *Crithidia bombi* growth and central metabolite content in the gut lumen. Although the *Bombus*-*Crithidia* system has been extensively studied [70], understanding such relationships may be the missing piece of the puzzle to fully grasp the *Bombus*-*Crithida* dynamic.

### 5.2. Giving Parasites a Chance

Parasites are often viewed as negative for pollinator health, even being pinpointed as one of the main factors of pollinator decline in the literature e.g., [71,72]. As a consequence, a growing number of studies have tried to find ways to control parasite infection in bees, notably through natural medicines found in floral resources [73]. However, as recently emphasized by Brown [74], while parasites negatively impact the health of pollinators *sensu stricto* (i.e., at the individual level), they enhance the health of pollinators *sensu lato* (i.e., at the community level). For instance, by limiting the populations of dominant generalist pollinators, parasites enable the survival and success of their rarer competitors [75]. Besides, infections have evolved a regulatory relationship with pollinator immunity—by being exposed to a low-virulent but abundant parasite, pollinators are protected against subsequent exposure to a high-virulent parasite (i.e., immune priming [76,77]). In fact, parasite communities *per se* are not responsible for pollinator decline but human-disturbed natural pollinator–pathogen dynamics are [78]. One may therefore argue the relevance of implementing floral enhancement schemes to reduce parasite prevalence in non-disturbed landscapes, whereas access to medicinal resources could significantly support bee populations in man-modified landscapes [73,79]. Self-medication should then be extended to other environmental stressors, mainly driven by human activities.

### 5.3. Mixing Behaviour, Narrow Diet and Self-Medication

One condition for self-medication to occur is that feeding on medicating resources should be detrimental to uninfected individuals [11,25]. Such detrimental effects may be due to (i) the excessive consumption of pharmaceutically active compounds, which often demonstrated little or no medicinal effects (i.e., hormetic effect [46,80]), or (ii) the poor nutritive value of medicating resources compared to nutritionally optimal resources [81]. In fact, although self-medication is used by organisms to reduce the costs of other strategies (e.g., activation of immune defense during infection), it does not mean that self-medication is devoid of any costs, especially if medicating resources are of poor nutritive values. In that case, the benefits of ingesting medicating resources are expected to exceed those of ingesting highly nutritive resources in infected organisms. In generalist bees, a trade-off between medicinal and nutritive benefits may be reached by mixing pollen from several plant species during foraging and within the nest [82]. For bee species displaying such mixing behaviours such as *B. terrestris*, feeding plasticity is then likely to be echoed by a change in the relative proportions of medicating and nutritive resources in infected individuals (i.e., increasing the proportion of therapeutic resources), rather than by an exclusive switch to medicating resources. This mixing strategy should therefore be considered when trying to detect self-medicating behaviours, even if it makes it less obvious for observers. As such a feeding plasticity is unexpected to occur in specialist species, the concept of self-medication could be limited to generalist species. However, it is also plausible that some medicating resources are actually highly nutritive, allowing specialization on one resource with both nutritive and medicating benefits, but it should be quite rare. In addition, specialist organisms could ingest or harvest non-food items for medicinal purpose, as already highlighted in social species (i.e., plant material or resin for nest construction, [20]). Hitherto, investigation of self-medication has been limited to generalist bee species, and mainly considered through the lens of food items.

## Figures and Tables

**Figure 1 biology-12-00497-f001:**
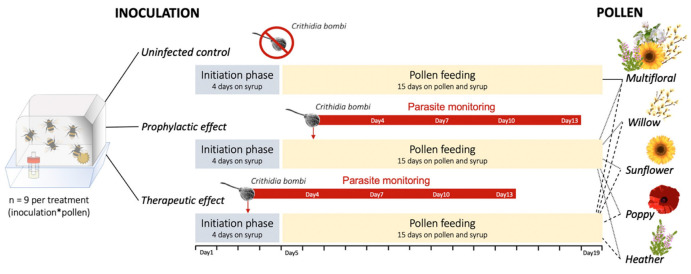
Experimental design used for testing prophylactic and therapeutic effects of five different pollen diets. The experiment timeline including parasite inoculation (red arrow) and monitoring is presented. Uninfected microcolonies fed multifloral pollen diet served as control during the experiment.

**Figure 2 biology-12-00497-f002:**
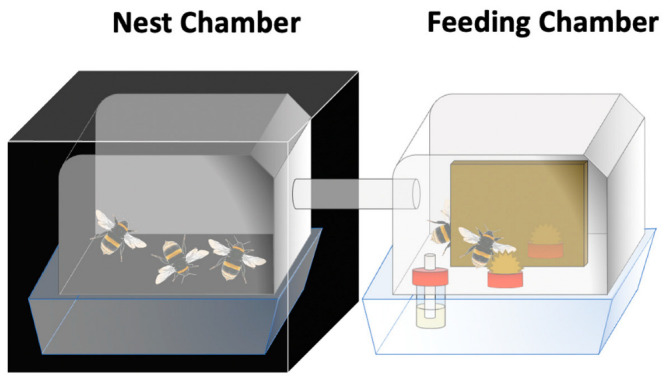
Experimental setup (i.e., two-box system) used for the feeding choice experiment. Nest chamber was kept into the dark with one small willow candy for brood initiation; feeding chamber was provided with syrup, kept under 12 h natural light conditions and divided in two compartments for worker pollen choice. Workers were enabled to freely move between the two chambers through a tube connection. Three pollen combinations were used namely, heather/multifloral, sunflower/multifloral, and heather/sunflower. All combinations were tested with infected and uninfected microcolonies. Nine microcolonies were used per treatment (infection*pollen combination).

**Figure 3 biology-12-00497-f003:**
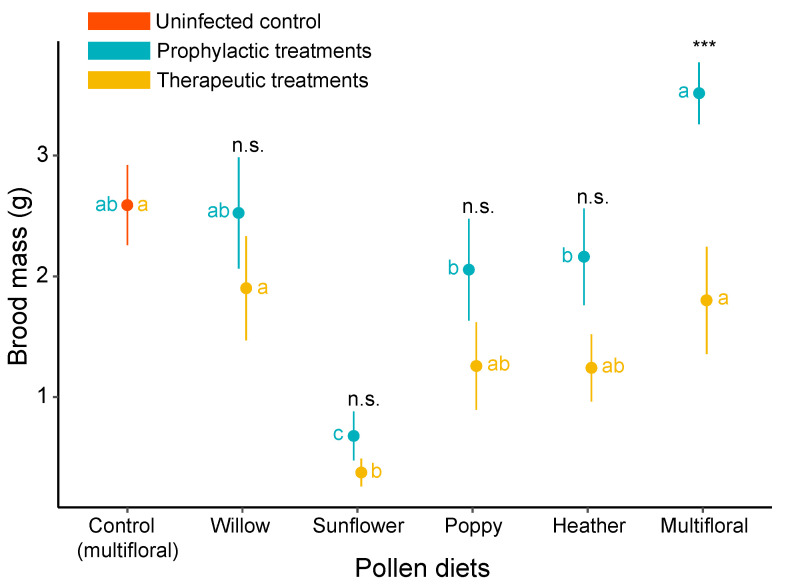
Effects of treatments on the total mass of hatched offspring in *B. terrestris* micro-colonies. Data in blue represent the prophylactic treatments, data in gold represent the therapeutic treatments, and data in orange represent the uninfected control (i.e., uninfected microcolonies fed on multifloral pollen). Points are mean values of each treatment and error bars indicate the standard error of means. Letters indicate significance at *p* < 0.05; blue letters refer to comparison across pollen diets within prophylactic treatments, gold letters refer to comparisons across pollen diets within therapeutic treatments, and statistical significance code refers to comparison between treatments (prophylactic vs. therapeutic) (n.s., *p* > 0.05; ***, *p* < 0.001). Pairwise comparisons were performed based on a priori contrasts.

**Figure 4 biology-12-00497-f004:**
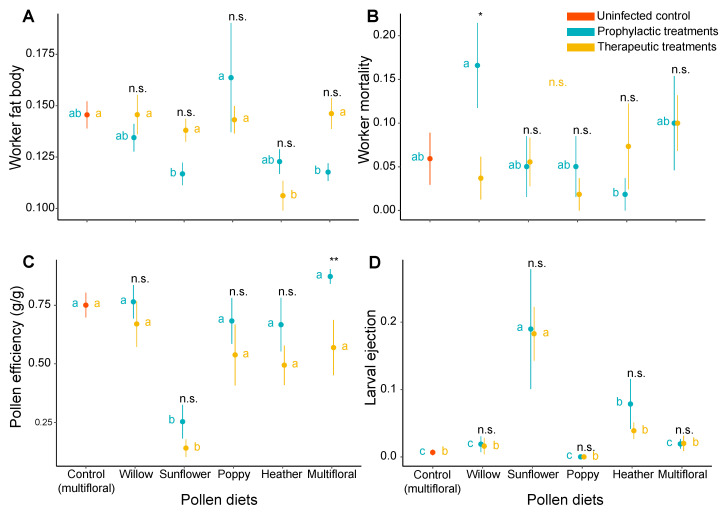
Effects of treatments on *B. terrestris* micro-colonies. (**A**) Worker fat body, (**B**) Worker mortality, (**C**) Pollen efficiency, and (**D**) Larval ejection. Data in blue represent prophylactic treatments, data in gold represent therapeutic treatments, and data in orange represent the uninfected control (i.e., uninfected microcolonies fed multifloral pollen). Points are mean values of each treatment and error bars indicate the standard error of means. Letters indicate significance at *p* < 0.05; blue letters refer to comparison across pollen within prophylactic treatments, gold letters refer to comparisons across pollen within therapeutic treatments, and statistical significance code refers to comparisons between treatments (prophylactic vs. therapeutic) (n.s., *p* > 0.05; *, *p* < 0.05; **, *p* < 0.01). Pairwise comparisons were performed based on a priori contrasts.

**Figure 5 biology-12-00497-f005:**
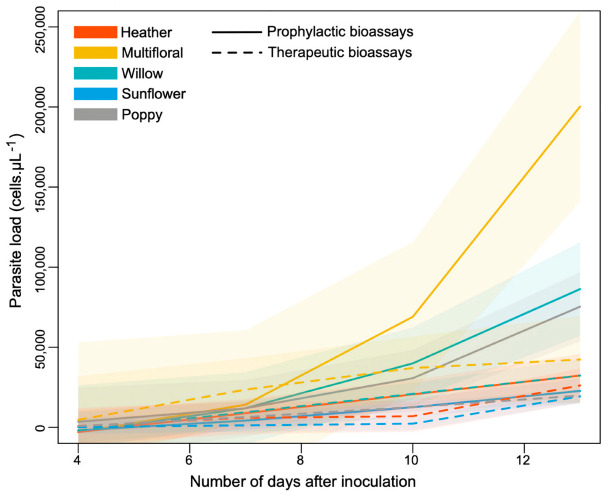
Dynamics of infection within *B. terrestris* microcolonies. The color code refers to the pollen diet, and the line type refers to the bioassay categories. Generalized additive mixed effect models (GAMMs) were used to fit smoothers to the data showing mean trends (±95% confidence intervals) over time.

**Figure 6 biology-12-00497-f006:**
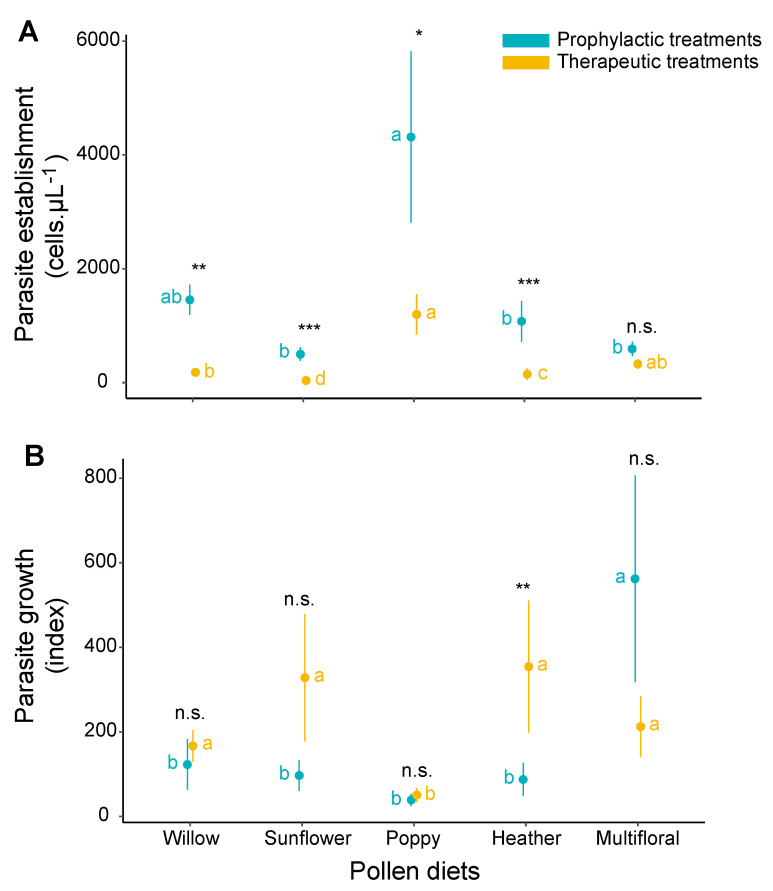
Effects of treatments on host tolerance in *B. terrestris* microcolonies. (**A**) Effect on parasite establishment (i.e., three days after inoculation), (**B**) Effect on parasite growth. Data in blue represent prophylactic treatments, and data in gold represent therapeutic treatments. Points are mean values of each treatment and error bars indicate the standard error of means. Letters indicate significance at *p* < 0.05; blue letters refer to comparisons across pollen within prophylactic treatments, gold letters refer to comparisons across pollen within therapeutic treatments, and statistical significance code refers to comparison between treatments (prophylactic vs. therapeutic) (n.s., *p* > 0.05; *, *p* < 0.05; **, *p* < 0.05; ***, *p* < 0.001). Pairwise comparisons were performed based on a priori contrasts.

**Figure 7 biology-12-00497-f007:**
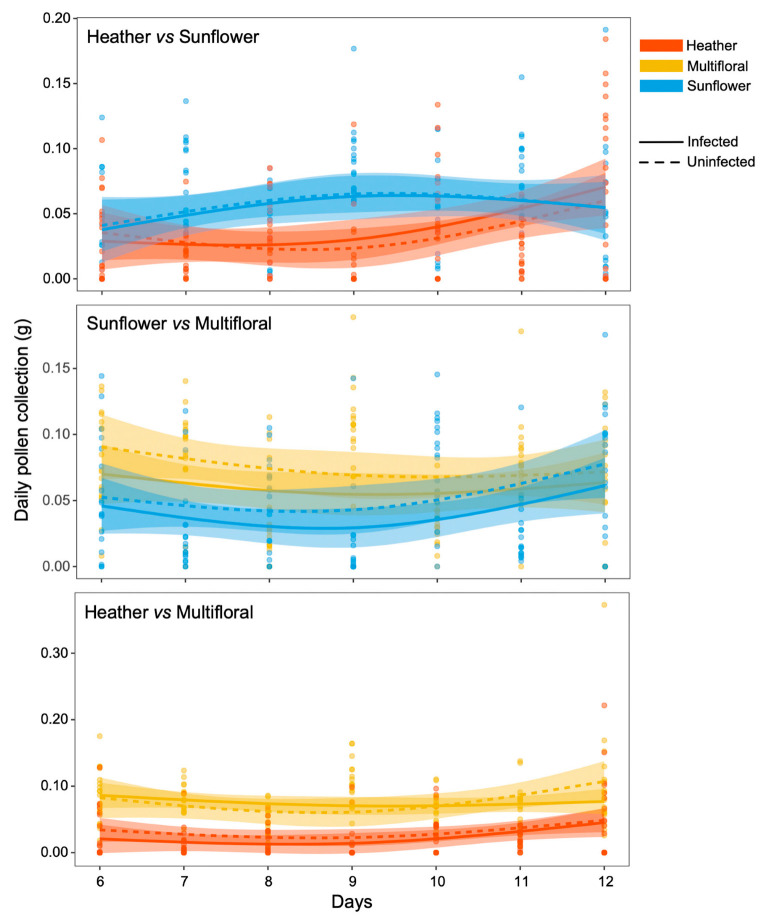
Amount of daily collected pollen (g, standardised by total mass of *Bombus terrestris* workers) in microcolonies being offered a choice between heather or sunflower pollen, sunflower or multifloral pollen, and heather or multifloral pollen. Generalised additive mixed effect models (GAMMs) were used to fit smoothers to the data showing mean trends (±95% confidence intervals) over time. Each dot represents one data point (i.e., standardised amount of collected pollen per day for each microcolony).

**Figure 8 biology-12-00497-f008:**
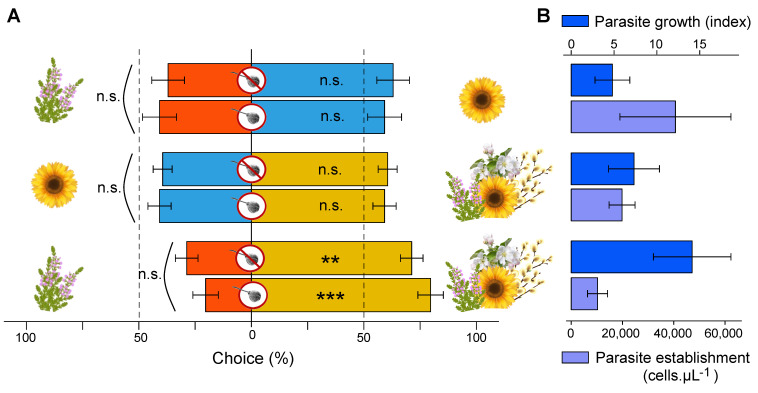
Self-medication behaviour in *B. terrestris* microcolonies. (**A**) Pollen preference across treatments (i.e., choice between two different pollen species in each treatment), and (**B**) effects of treatments on parasite establishment (i.e., three days after inoculation) and growth (n.s., *p*  >  0.05; **, *p*  <  0.01; ***, *p*  <  0.001).

## Data Availability

Upon acceptance, datasets generated and analysed during the current study will be provided via the figshare repository (Experiment 1: 10.6084/m9.figshare.21941318; Experiment 2: 10.6084/m9.figshare.21941315).

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
