# Peer review of "Pollen as Bee Medicine: Is Prevention Better than Cure?"

_biology, 2023, doi:10.3390/biology12040497_

Round 1
Reviewer 1 Report
Summary
This study investigates the putative therapeutic and prophylactic effect of different pollen diets on bumble bee health. The bumble bees were infected with the parasite Crithidia bombi. Different health parameters were determined to examine host tolerance and host resistance in one experiment and self-medicative behavior in another experiment. The topic of this study is interesting and to my knowledge there are not many studies investigating self-medication in insects yet. I recommend the major revision of the manuscript. I have addressed my major concerns below which need to be addressed.
Major comments:
- Line 129-131: Why is red poppy pollen not included in the multifloral diet but the other three potentially medicative pollens?
- Line 162-164: Did the replacement workers receive the same treatments as the according treatment group (infection vs. no infection with prophylactic vs. therapeutic pollen) before they were put into the running experiment? If mortality is determined, the substitution with fresh workers is problematic because how to discriminate between old and fresh workers.
- Line 215-216: What if the one marked worker died during the experiment? What was the actual number (n) of examined workers per treatment?
- Line 228-229: Why was the time point at day 4 (i.e., 3 days post-inoculation) chosen for parasite establishment? How do you know that this is the time when the parasite is established?
- Line 260-261: Dead workers were removed but not replaced here. Why was this done differently than in experiment 1? In experiment 1 pollen collection data was measured as well.
- Line 295-296: Why does the uninfected control belong to the therapeutic treatments? Also, according to Fig. 3, in the therapeutic treatments, the groups which received willow, multifloral, poppy, or heather were not significantly different from the control. Only the group which received sunflower diet had a significantly lower brood mass than the control. The sentence is confusing and should be rephrased.
- Line 328 ff., Figure 3: The control group also received a multifloral diet. Thus, the control point with error bars should be together with the multifloral category instead of being a category itself. This is also the case in Fig. S1.
- Line 366-369: Please add “(Fig. 4)” at the end of this sentence.
- Line 377, ff.: Is Fig. S2 correct here? The descriptions of the results of parasite load and parasite establishment seem mixed in this paragraph. It is hard to comprehend which parameter (parasite load or parasite establishment) the statistical values and sentences are about. Please make this paragraph clearer.
- Lines 459-473: The discussion about mortality can be omitted as there was not a significant difference between the infected groups and the control. Furthermore, as dead bumblebee workers were replaced with new ones, the evaluation of mortality is problematic anyway. Moreover, if there was a bias due to pre-evolved C. bombi in regards to willow, the experimental set-up should be reconsidered and the data on willow-pollen left out because the treatment groups with willow pollen then do not compare to the rest. Please consider that the mortality in the group with prophylactically-fed willow pollen is not significantly different from the uninfected control.
- Lines 499-502: Have later time points be examined than Day 4? For example, it would be interesting to compare to the situation at the last day of the experiment (Day 13).
- Lines 502-503: According to Fig. 4, no plateau is reached in prophylactic treatments with a multifloral, willow and sunflower diet.
- Line 563: This paragraph about future directions is very lengthy and theoretical. Please shorten.
- Line 672: How was Crithidia bombi identified (method)?
Supplementary Materials:
- First, I do not understand why the Figures S1, S2, and S3 are in the supplementary materials while the tables 1, and 2 are in the main manuscript. I find the figures much more informative and would rather put the figures in the main text and the tables in the supplementary materials.
- Fig. S1:
o The units for worker fat body (Fig. S1A), worker mortality (Fig. S1B), and larval ejection (Fig. S1D) are missing.
o The gold letters for therapeutic treatments in worker mortality (Fig. S1B) are missing.
Minor comments:
- Line 413: prefer instead of preferred
- Supplemetary Materials: Please check the format of the header and the size of Figure S1.
Author Response
Dear Reviewer,
Thank you very much to allow us to resubmit our manuscript (biology-2208199). We are grateful for the time and effort that you have invested in reading the manuscript and making suggestions about ways in which we could improve the text. The manuscript has been revised to take account of the suggestions made. In the attached document, we indicate how we have responded to your comments and suggestions (each of our responses starting with “--”). The line numbers refer to the modified manuscript without marks.
Kind regards,
Maryse Vanderplanck (on behalf of all authors)

Reviewer 2 Report
Simple Summary and Abstracts:
More results should be given at the expense of introductory sentences.
Discussion:
The authors should be more careful when discussing the results related to the influence of pollen diet on Crithidia-infected bumblebees, considering that in their experiment critidia did not cause the supposed harmful effects of Crithidia infection on "microcolony growth, individual immunocompetence or worker behaviour, as expected" (as they claim in lines 434-436).
Under the subchapter 4.2.1. "The need for virulent parasites", the wrote that the "did not observe any detrimental effect of Crihidia on bumble bee fitness" (line 538) and soon after that wrote that "no therapeutic effect on host resistance has been observed in bumble bee workers that were offered the two medicinal resources" (543-544). So, I am wondering what therapeutic effects may be expected in hosts that are not affected by infection?
I also inserted some suggestions directly in PDF (attached).

Author Response

(The authors gave the same response as above.)

Reviewer 3 Report
The authors presented an extensive work on the medicinal effects of different pollens on bumblebees infected by Crithidia bombi. I think it is a very interesting work, especially because, as the authors point out, it does not only focus on forced-feeding experiments.
All sections of the manuscript are written clearly and comprehensively. The results section is a bit heavy to read, especially when statistical values are listed, but on the other hand there were many treatments and comparisons to describe. So I do not know if there would be a better way to organise this part. I found no language errors.
Therefore, in my opinion, the manuscript can be accepted for publication in the present form.
Author Response

(The authors gave the same response as above.)
